# Skeletal Ni electrode-catalyzed C-O cleavage of diaryl ethers entails direct elimination via benzyne intermediates

Yuting Zhou [1✉], Grace E. Klinger[1,2], Eric L. Hegg [2], Christopher M. Saffron [3,4] & James E. Jackson [1✉]

Diaryl ethers undergo electrocatalytic hydrogenolysis (ECH) over skeletal Ni cathodes in a mild, aqueous process that achieves direct C-O cleavage without initial benzene ring saturation. Mechanistic studies find that aryl phenyl ethers with a single para or meta functional group (methyl, methoxy, or hydroxy) are selectively cleaved to the substituted benzene and phenol, in contrast to recently reported homogeneous catalytic cleavage processes. Ortho positioning of substituents reverses this C-O bond selectivity, except for the 2-phenoxyphenol case. Together with isotope labeling and co-solvent studies, these results point to two distinct cleavage mechanisms: (a) dual-ring coordination and C-H activation, leading to vicinal elimination to form phenol and a surface-bound aryne intermediate which is then hydrogenated and released as the arene; and (b) surface binding in keto form by the phenolic ring of the hydroxy-substituted substrates, followed by direct displacement of the departing phenol. Notably, acetone inhibits the well-known reduction of phenol to cyclohexanol, affording control of product ring saturation. A byproduct of this work is the discovery that the ECH treatment completely defluorinates substrates bearing aromatic C-F and C-CF$_3$ groupings.

[1] Department of Chemistry, Michigan State University, East Lansing, MI 48824, USA. [2] Department of Biochemistry and Molecular Biology, Michigan State University, East Lansing, MI 48824, USA. [3] Department of Biosystems and Agricultural Engineering, Michigan State University, East Lansing, MI 48824, USA. [4] Department of Chemical Engineering and Material Science, Michigan State University, East Lansing, MI 48824, USA. ✉email: zhouyut2@msu.edu; jackson@chemistry.msu.edu

Catalytic methods to cleave diaryl ether (DAE) bonds are needed (a) to break down plant matter, with its DAE-rich lignin, into renewable replacements for fossil carbon building blocks; and (b) to mitigate pollution from modern agrochemicals that include persistent DAE moieties[1–3]. Despite this relevance to sustainability and environmental issues, relatively few reports of effective and selective DAE degradation strategies have appeared.

To address global warming concerns due to the rise in fossil fuel-derived atmospheric carbon dioxide[4–7], new tools are being sought to enable a switch from fossil-to renewable carbon (biomass) for the production of chemicals and liquid fuels. Most efforts to date have focused on the largest component of biomass: carbohydrates (cellulose and hemicellulose)[8]. Lignin, the other major fraction[9,10], has potential as a carbon-rich feedstock to make both renewable fuels and chemicals[11–13], but its diverse, strong linkages make it recalcitrant. To cleave lignin's most common linkage, the $sp^3$ C-O bond of the β-O-4 aryl alkyl ether (Fig. 1a, highlighted in blue), various strategies have been devised, such as oxidatively activated reductive cleavage[14–18], protectionreduction schemes[19–21], and photochemical degradation[22,23]. But the second most common linkage, the 4-O-5 ether (Fig. 1a, highlighted in red) has received less attention, perhaps because its diaryl ether C-O bonds are among the strongest in the lignin polymer (c.f. the 86 kcal/mol C-O bond dissociation energy of diphenyl ether)[24].

Diaryl ether substructures are also important in recently developed agrochemicals and some drugs[25], endowing them with outstanding hydrophobicity, lipophilicity, cell wall penetration, and resistance to degradation[25,26]. However, unlike their hydrophilic sidechains, the DAE headgroups in these products resist environmental breakdown (Fig. 1b)[27], accumulating as persistent organic pollutants (POPs) in surface and ground waters[3,28–30]. For example, of the reported 25 environmental fates of tolfenpyrad, an insecticide recently approved for usage in the US, none successfully breaks down the DAE moiety[30–32].

Some homogeneous catalytic DAE hydrogenolysis reactions show good selectivity and reduction control. However, these methods typically involve specialized ligands and may also require unique precursors for the catalytic metal centers. Those so far reported also require inert organic solvents, not readily compatible with the non-inert natural environments of the above described DAE substrates[33–37]. On the other hand, their molecular, homogeneous conditions typically allow detailed mechanistic studies.

Heterogeneous catalysis offers potentially greater compatibility with the complexity of biomass, especially the associated water. However, to cleave the strong $sp^2$ C-O bonds in DAEs, classic hydrogenation methods commonly run at high pressure (40–65 bar) and temperature (200–300 °C) reaction conditions[38–40]. Some milder hydrogenolyses have been developed, mostly requiring precious metals (Pd[41], Rh[42], Ru[41,42], Pt[43]), but improved conditions (120–150 °C; 6–12 bar $H_2$) with Ni nanoparticles have recently been reported[44,45]. Nonetheless, two issues remain: (a) these robust but severe heterogeneous catalytic DAE cleavage reactions show low selectivity, with ring saturation of the phenolic products hard to avoid; and (b) most reports have focused on diphenyl ether (DPE) alone, whereas the DAE scaffolds noted above carry various functionalities[25].

Herein, we explore the reactivities and C-O bond selectivities in the cleavage of substituted DAEs via electrocatalytic hydrogenolysis/hydrogenation (ECH). A diverse slate of DAE substrates is subjected to aqueous phase ECH with earth-abundant skeletal Ni cathodes at 60 °C and ambient pressure, as shown in Fig. 1c. This work builds on that of Lessard et al. and on our own background in the development of electrocatalytic organic reductions over skeletal nickel catalysts[46–49]. By forming the active hydrogen species by proton reduction on the catalyst surface (reflected by bubbles evolving from the cathode), ECH avoids the hazards of handling $H_2$ gas, the limitations of $H_2$ solubility and transport in aqueous solvent, and the kinetic barrier to $H_2$ dissociation. However, the structural complexity of the skeletal metal and the three-phase (solid, liquid, gas) nature of phenomena there prevent the use of surface spectroscopies or credible quantum chemical simulations[49,50] to probe substrate-catalyst interactions. Consequently, our mechanistic investigations center on substituent effects beyond the parent DPE case[44,51], reaction rates, selectivities, isotopic labeling, and solvent and inhibition studies to understand the aryl ether C-O reductive electrocatalytic hydrogenolysis processes.

## Results

**Overview.** Initial studies of the parent DPE revealed that ether cleavage occurs prior to phenyl ring hydrogenation. Comparisons were then pursued among four functionalized DAEs—diphenyl ether, 4-phenoxyanisole, 4-phenoxytoluene and 4-phenoxyphenol —together with their reaction products. Based on the combined results of rate comparisons, substituent electronic and positional studies, isotope labeling experiments, and regioselectivity investigations, two distinct mechanisms for C-O cleavage were established for the four representative functionalized substrates. Extension to ortho- and meta-substituted analogues, and analysis of the effects of a suite of additional substituents revealed their impacts on the rates and selectivities of aryl C-O bond cleavage, and also uncovered the activation of various functional groups

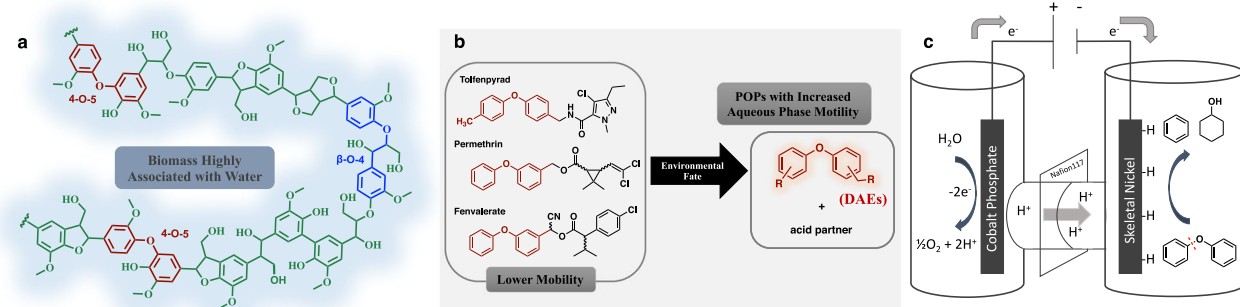

**Fig. 1 Occurrence of diaryl ethers and schematic representation of their cleavage via electrocatalytic hydrogenolysis (ECH). a** Lignin structure, highlighting diphenyl ether linkage (4-O-5). **b** POPs with increased mobility from environmental degradation of widely used agrochemicals. **c** Schematic illustration of electrocatalytic hydrogenolysis/hydrogenation of diphenyl ether in an H-cell.

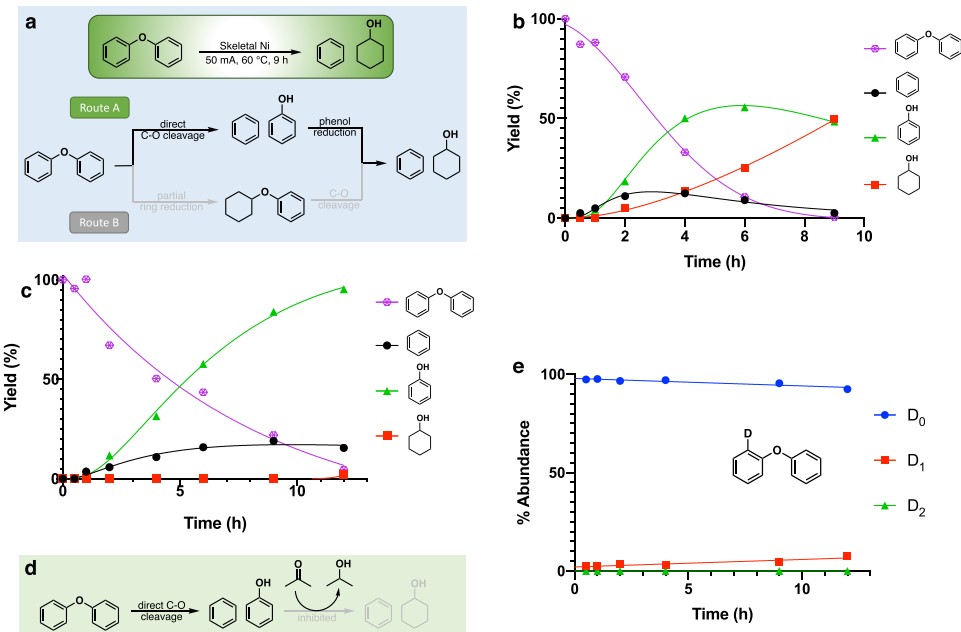

**Fig. 2 Determination of the major cleavage pathway for diphenyl ether. a** Two cleavage pathways of diphenyl ether. **b** Electrocatalytic hydrogenation (ECH) of diphenyl ether (8.33 mM) over a skeletal nickel electrode at 50 mA (current density 8 mA/cm$^2$) in the 30 mL of 0.1 M electrolyte mixture (2:1 pH 8 borate buffer:IPA) at 60 °C under ambient pressure (Standard Conditions). **c** ECH of diphenyl ether under standard conditions in mixture of buffer/IPA/acetone (40:17:3). **d** Ring reduction inhibited by acetone reduction. **e** H/D isotope labeling of diphenyl ether with acetone treated skeletal Ni.

(CH$_3$, CF$_3$) commonly seen as unreactive. The mechanistic insights obtained also uncovered conditions in which the strong sp$^2$ C-O ether bond could be cleaved without over reduction of the monomeric phenolic products.

**The C-O bond in diphenyl ether is directly cleaved without ring hydrogenation**. Electrocatalytic hydrogenolysis (ECH) of DPE produced benzene and cyclohexanol (Fig. 2a). Two potential routes to these products can be envisioned for the cleavage process. In Route A, DPE is directly cleaved to benzene and phenol. Phenol then undergoes fast ring saturation to generate cyclohexanol. In Route B, one of the phenyl rings first undergoes ring hydrogenation, converting it into cyclohexyl phenyl ether; C-O cleavage then gives benzene and cyclohexanol.

To determine which of paths A or B is preferred, the reaction time sequences were analyzed. As shown in Fig. 2b, ECH of DPE formed benzene and phenol, with a slower buildup of cyclohexanol. The high hydrophobicity and high vapor pressure of benzene made it difficult to retain in the 60 °C polar aqueous buffer (for details, see Supplementary Fig. 1). Importantly, however, neither cyclohexane nor cyclohexyl phenyl ether were detected. If the latter aryl alkyl ether product were indeed formed, it would not be expected to cleave to cyclohexanol and benzene; other simple alkyl phenyl ethers such as anisole are unreactive under these conditions. These findings rule out route B, consistent with reports from Lercher and co-workers studying more classical catalytic hydrogenation/hydrogenolysis conditions[44].

In a further test confirming the dominance of route A, a small amount (5%) of acetone was used as a co-solvent in the ECH of DPE under otherwise standard conditions (Fig. 2c). Our previous study of 2-phenoxyacetophenone cleavage had found acetone to be an inhibitor of phenol ring reduction and explored its use to tune reduction selectivity[49]. As shown in Fig. 2d, addition of 5% acetone almost completely inhibited phenol reduction, with barely any cyclohexanol seen after 12 h of continuous ECH at

50 mA. Thus, the direct cleavage pathway A is preferred; if partial or total ring reduction (Route B) preceded ether cleavage, in the presence of acetone as co-solvent, cyclohexanol should arise in parallel with phenol production.

**Diphenyl ether undergoes slow isotope exchange**. To further probe the cleavage mechanism of DPE, isotope labeling experiments were performed. H/D exchange was conducted at 60 °C in a 2:1 mixture of D$_2$O and nondeuterated 2-propanol (IPA). As shown in Fig. 2e, the amount of exchange observed in DPE with discharged nickel catalyst after 12 h was only ~8%. The exchange occurred ortho to the ether C-O bond, as shown by $^1$H NMR (see Supplementary Fig. 6). Under more vigorous reaction conditions using freshly activated Ni catalyst with applied current, the amount of isotope incorporation was still very low in the recovered DPE, as it was undergoing cleavage into benzene and phenol (for details, see Supplementary Fig. 8a). This suggests that upon Ni insertion into the ortho C-H bond of the activated benzene ring, forward reaction to cleave the C-O ether bond dominates over reversal that would scramble the ortho hydrogens with surface deuterium atoms.

The location of the H/D exchange suggested the possibility that ether bond cleavage might proceed via ipso-ortho addition of hydrogen and phenol elimination. As shown by the studies above, however, the activated ring of diphenyl ether is not reduced prior to the C-O cleavage. A more attractive alternative exchange pathway is reversible ortho-metallation of a C-H site, pointing to a benzyne pathway for ether cleavage (vide infra).

**Proposed C-O cleavage mechanisms of diphenyl ether**. On the basis of the above information, three different direct C-O cleavage mechanisms of diphenyl ether were proposed. In route I, double ring coordination allows ortho C-H insertion on the phenyl ring that is perpendicular to the catalytic surface (Fig. 3a, Route I). Elimination of the vicinal phenoxide then forms a surface-bound benzyne intermediate, which is rapidly reduced to benzene by

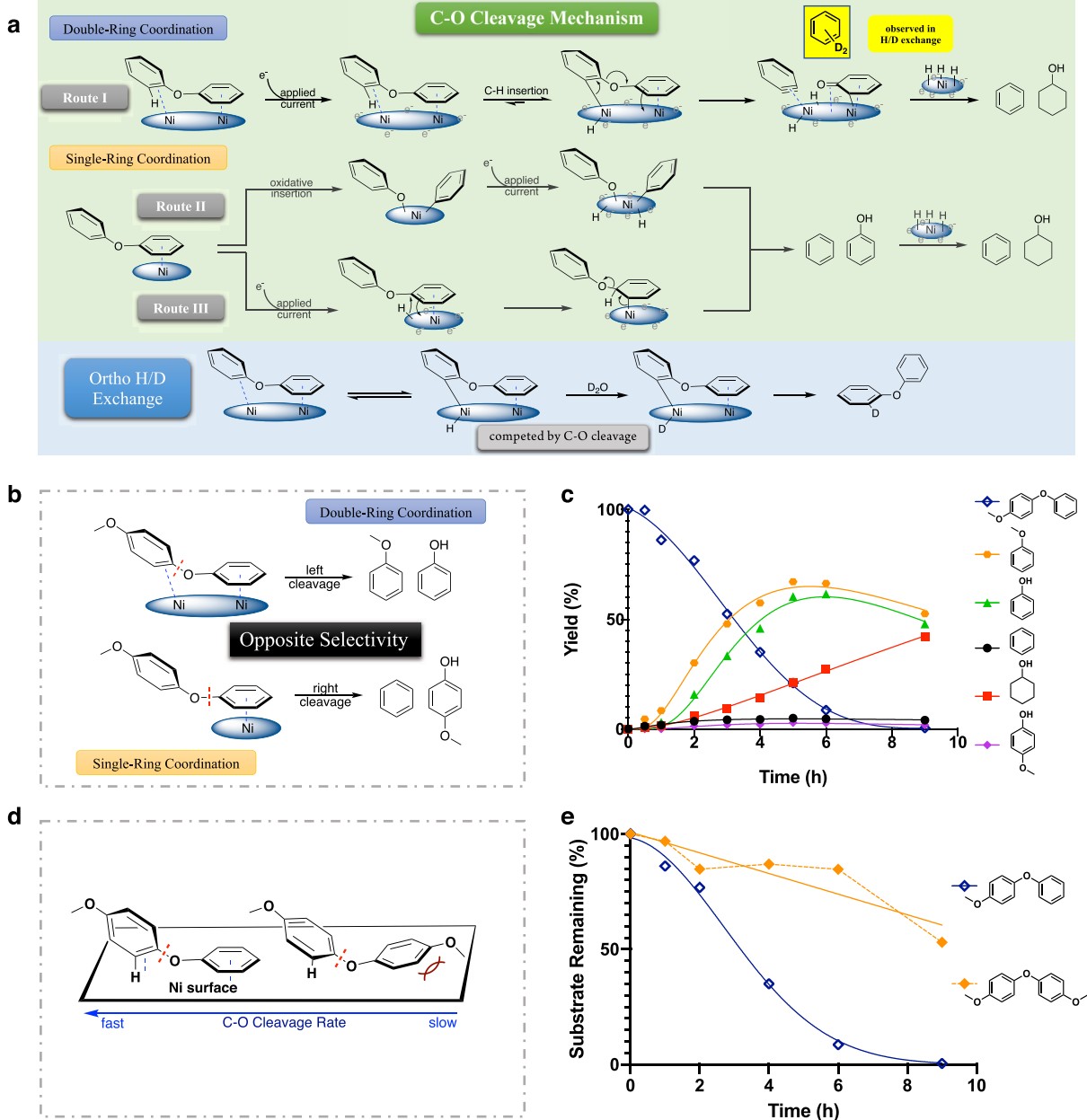

**Fig. 3 Identification of the major C-O cleavage mechanism for diphenyl ether. a** Proposed C-O cleavage and H/D exchange mechanisms of diphenyl ether. **b** Asymmetric diphenyl ethers differentiate the proposed cleavage mechanisms. **c** ECH of para-methoxylated diphenyl ether over skeletal nickel electrode under standard condition (50 mA, 60 °C, 33% v/v IPA). For the initial substrate, "yield" represents the percentage of starting material recovered at each time point. **d** Double ring coordination requires both perpendicular and parallel bindings. **e** Reactivity comparison between 4-phenoxyanisole and bis-4-methoxyphenyl ether. For additional time courses and product analyses see supplementary Fig. 69. The dotted orange line highlights the variability in analysis due to borderline solubility of this substrate in the reaction medium.

reaction with surface hydrogen atoms. This double-ring coordination mechanism was proposed based on the appearance of di-deuterated benzene in the cleavage products formed in the H/D exchange experiments (for details, see Supplementary Fig. 3). It explains not only the fast reduction of the phenol intermediates, but also the label distribution in the product, and the reaction's sensitivity to substituents (vide infra).

An alternative candidate cleavage via single ring activation is shown in route II (Fig. 3a). Here, with one of the phenyl rings coordinated to the catalyst, oxidative C-O insertion by Ni followed by reductive elimination with hydrogen could directly break the ether bond. This pathway has been demonstrated in single metal (Ni) molecular catalysis studies, where the nickel

center was spatially positioned and electronically modified by its ligand donor set[37]. Lastly, Route III, also beginning with single ring binding, would activate the phenyl ring by delivery of a surface-bound hydrogen atom to the ipso sp[2] carbon of the bound ring, followed by rearomatization of the coordinated ring via cleavage of the C-O bond. As presented in Fig. 3a, the major differences between routes I and II-III are at the ring activation step. To identify the preferred DAE cleavage mechanism, further time-course, substituent effect, and isotopic labeling studies were pursued.

Besides the proposed cleavage paths, Fig. 3a also outlines a mechanism for the H/D exchange in DPE. In contrast to the fast C-O cleavage, the isotope incorporation is slow. Evidently, if the

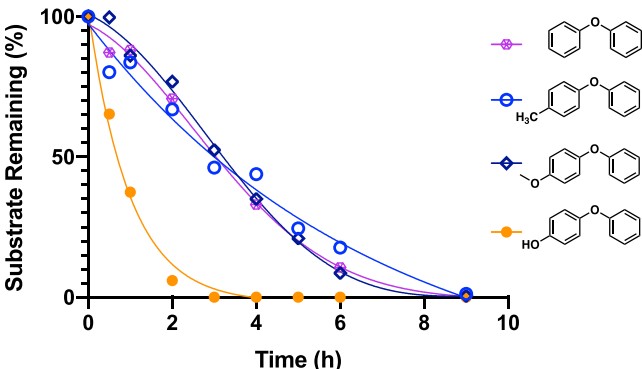

**Fig. 4 Substituent effects on DAE cleavage rates.** C-O ether bond cleavage rate of different electron-donating group substituted diphenyl ether under standard ECH condition. For additional time courses and product analyses see supplementary Fig. 9.

ortho carbon activation represents the first step in the cleavage process, the forward C-O cleavage predominates over the reverse process, replacing the original hydrogen with deuterium. For this overall hydrogenolysis (DPE + H₂ → benzene + phenol; exothermic by 11.6 kcal/mol)[24], presumably the increase in entropy and in solvation upon phenoxide elimination further favors C-O cleavage. Alternative exchange paths such as reversible hydrogenation involving the ortho carbon appear less likely, given the observed resistance of aryl ethers to hydrogenation. Addition would also lead to an anti-relationship between the bound Ni surface and the ortho C-H in the CHD site, complicating the release of the ortho hydrogen and retention of deuterium.

**Cleavage selectivities in DAEs support route I as the main cleavage mechanism.** Proposed cleavage routes I-III differ primarily in the ring activation steps (double vs. single coordination). These predict opposite cleavage selectivities; the dual-ring coordination would lead to cleavage of the left C-O bond, while the single-ring coordination cleaves the right C-O bond. Study of the symmetrical diphenyl ether substrate could not fully differentiate the mechanisms, since all the paths lead to the same products (benzene, phenol). However, with a substituent on one of the aryl rings, the different coordination modes would generate different cleavage products (Fig. 3b). As noted in our previous report on β-O-4 models, the negatively charged Ni electrode is poor at coordinating and reacting with aromatic rings bearing electron-donating substituents such as the methoxy functionality[49]. Therefore, 4-phenoxyanisole, the singly para-methoxylated DPE was synthesized and its cleavage regioselectivity was examined. As shown in Fig. 3c, this substrate was cleaved completely within 7 h, giving anisole and phenol as the major cleavage products, consistent with the two-ring coordination mechanism (Fig. 3a, Route I). This cleavage result (anisole and phenol) is also opposite to those seen in previous reports interpreted in terms of oxidative insertion (Route II)[35,37]. Meanwhile, the reaction of bis-(4-methoxyphenyl) ether (disubstituted methoxylated model) was extremely sluggish (For details, see Supplementary Fig. 69). The rate contrast between mono- and di-substituted methoxylated DPE (Fig. 3d, e) also supports the two-ring mechanism. Further evidence for Route I is found in the H/D exchange results (vide infra).

Importantly, ECH of para-methoxylated DPE also rules out hydrolysis as a significant pathway of diaryl ether C-O cleavage; hydrolysis would yield 4-methoxyphenol and phenol. But as shown in Fig. 3c, anisole and phenol were the major cleavage products, the hydrolysis pathway could contribute at most <2%

(in fact, the minimal 2% of 4-methoxyphenol was matched with benzene co-product).

**Substituent effects on DAE cleavage rates.** As shown in Fig. 4, different substituents on the aromatic ring substantially impact the diaryl ether cleavage rates. As predicted by Route I, the cleavage rates of plain DPE (purple hexagon) and the para-methoxylated (navy diamond) and methylated (blue hollow circle) analogues are similar, suggesting that in the cleavage mechanism these moieties do not directly interact with the catalytic surface in a way that significantly affects the cleavage processes (Fig. 3b).

A strikingly different result was observed in the case of 4-phenoxyphenol, the singly para-hydroxylated analogue (orange circle) of DPE. Here, the presence of the hydroxyl group in the aromatic ring significantly accelerates the C-O cleavage. Our previous studies suggested that the electron-rich nucleophilic nickel cathode surface has a high affinity for the carbonyl functionality[49]. As with phenol itself, the hydroxyl site on the phenolic ring can equilibrate with the keto form, which binds tightly to the nucleophilic nickel surface, resulting in rapid cleavage.

**Reactivity and isotope exchange are similar in DPE and 4-methoxylated DPE.** As shown in Fig. 5a, b, the exchange locations of the differently substituted diphenyl ethers were very different. Like the parent DPE, the para methoxylated DPE (Fig. 5a) had slow deuterium exchange; only ~15% D was incorporated after 12 h of treatment at 60 °C. Thus, the DPE and 4-methoxylated DPE were similar in two ways: First, in terms of reactivity, these two substrates showed similar, fast C-O ether bond cleavage (Fig. 4) and slow H/D exchange (Figs. 2e, 5a). Second, the exchange locations were at the same carbon site for both, ortho to the C-O bond being cleaved. Notably, the exchange in the latter substrate occurs on the anisole ring, not the phenyl. Meanwhile, the anisole formed as the cleavage product was doubly deuterated at a level almost independent of the slow deuteration of starting material (for details see Supplementary Fig. 10). These similar behaviors imply that these two diaryl ethers react via similar cleavage mechanisms (Fig. 3a, Route I, and Fig. 5c).

**Fast benzylic activation on the methyl group is independent of the ether cleavage mechanism.** Introducing a methyl moiety to the diphenyl ether system slightly slowed the overall cleavage rate (Fig. 4). More importantly, it also presented additional H/D exchange behavior. As shown in Fig. 5b, within 12 h the methyl group showed nearly complete deuteration, as expected from benzylic C-H insertion by the deuterated nickel catalyst. Similar growth rates of D₁, D₂, and D₃ suggested that once the Ni inserted at the benzylic carbon, the incorporation of D happened rapidly (Fig. 5d).

The fast H/D scrambling does suggest easy and reversible binding of the methyl group to the catalytic surface; however, this binding does not accelerate the cleavage like the hydroxyl group does. Notably, it also does not activate ring H/D exchange ortho to the methyl substituent. Instead, as in the parent and para-methoxylated models, a small amount of H/D exchange was detected ortho to the C-O ether bond that was being cleaved (Fig. 5b, exchange locations confirmed by NMR spectra, see Supplementary Fig. 18). Thus, on the basis of its cleavage products (toluene and phenol; see Supplementary Fig. 9b), and this similar aromatic C-H activation, we infer that cleavage of para methylated DPE proceeds via the double ring coordination mechanism (Fig. 3a, Route I, and Fig. 5d) like the parent and methoxylated DPEs. As predicted by this picture, a small amount of penta-deuterated toluene was detected among cleavage products from the exchange experiment (for details, see

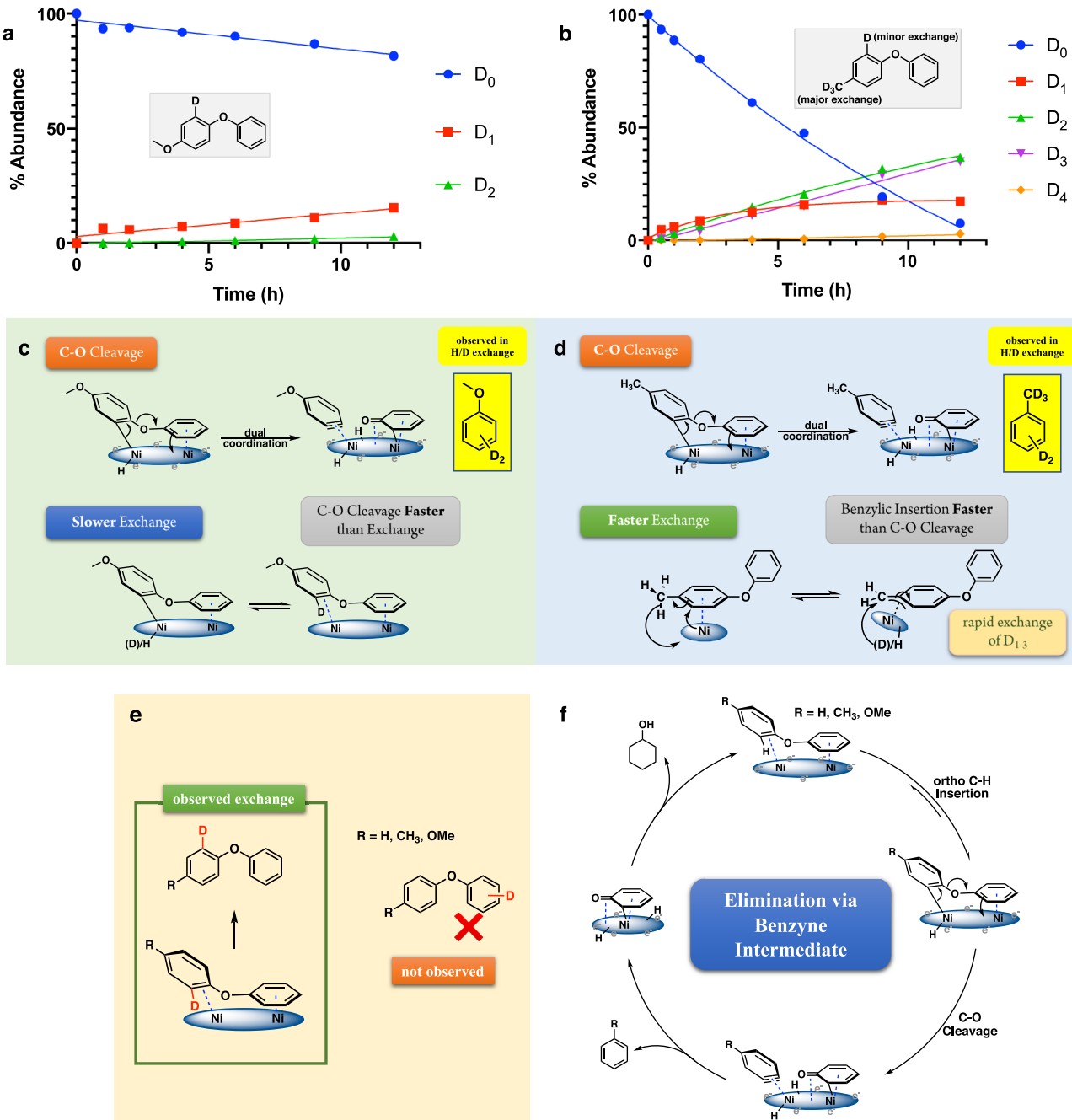

**Fig. 5 Effects of methyl and methoxy substituents on diaryl ether cleavage mechanism. a** H/D exchange time courses and exchange locations for methoxylated diphenyl ether. **b** H/D exchange time courses and exchange locations for methylated diphenyl ether. $D_n$ ($n = 0-4$) represent numbers of deuterium atoms incorporated. For additional analyses (NMR and MS Spectra) see supplementary Figs. 4–19. **c** Proposed cleavage and exchange mechanisms for methoxylated diphenyl ethers. **d** Proposed cleavage and exchange mechanisms for methylated diphenyl ethers. **e** Exchange at the ortho C-H of the substituted ring, consistent with the dual-ring coordination. **f** Proposed cleavage mechanism of DPE, methylated DPE and methoxylated DPE.

Supplementary Fig. 15), consistent with the proposed surface-bound methylbenzyne intermediate. Thus, the ether cleavage reaction proceeds via the same mechanism as the unsubstituted DPE (Route I), but with a small slowdown due to competition with the methyl group for binding. Analogous behavior is also seen in the trifluoromethyl-substituted analogue, as detailed below.

**Dual-ring coordination results in an ortho C-H activation on the substituted ring**. The location of the H/D exchange of the substituted DPEs (both methoxy and methyl cases), also

supports the dual-ring mechanism illustrated in Fig. 5e. Orienting the substituted ring perpendicular to the Ni surface enables the activation of this ortho aromatic C-H but not those of the unsubstituted phenyl ring. Meanwhile, the parallel adsorption of the unsubstituted phenyl ring to the nickel surface is also required. Notably, bis-(4-methoxyphenyl) ether has low reactivity (Fig. 3e, as well as Supplementary Fig. 69). Overall, a dual ring coordination mechanism that leads to a surface-bound aryne intermediate is proposed for the cleavage of diphenyl ether (DPE) and its corresponding methoxy and methyl congeners (Fig. 5f).

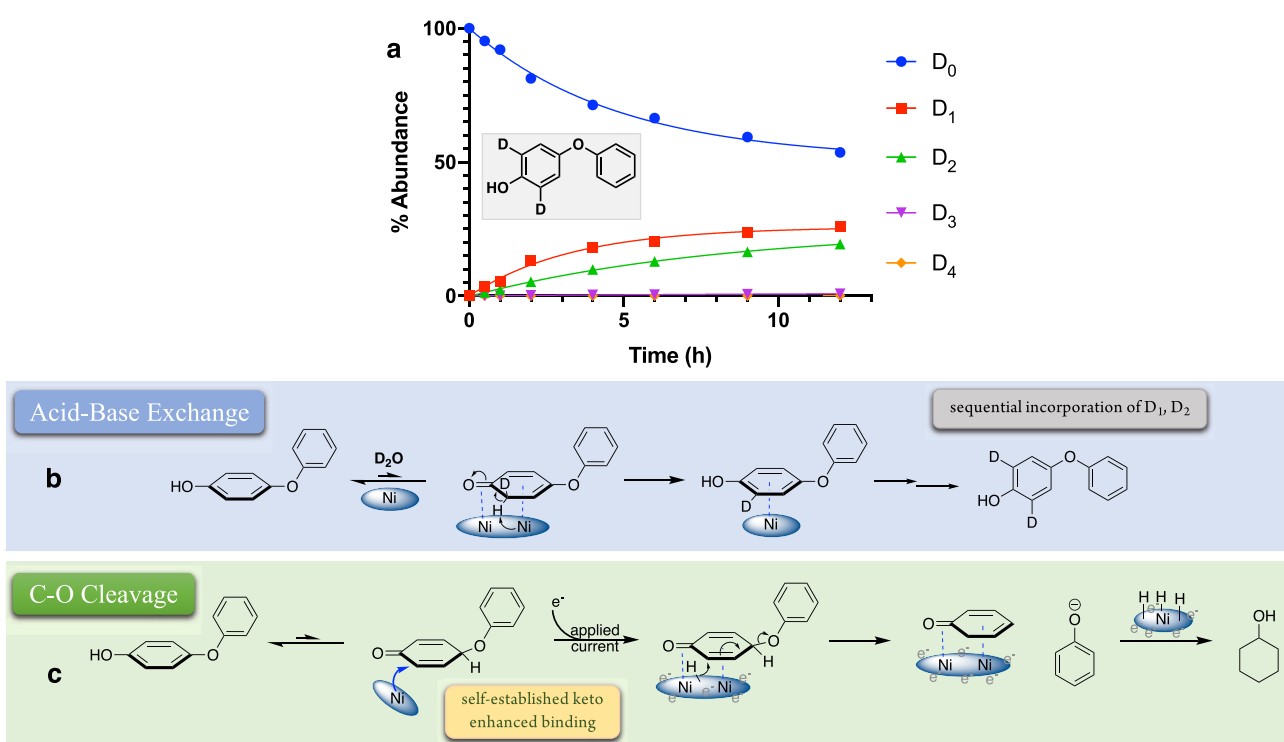

**Fig. 6 Effects of hydroxyl substituent on diaryl ether cleavage mechanism. a** H/D exchange time courses and exchange locations for hydroxylated diphenyl ether (analysis included a $H_2O$ wash to eliminate the rapidly exchangeable hydroxyl H). $D_n$ ($n = 0-4$) represent numbers of deuterium atoms incorporated. For additional analyses (NMR and MS Spectra) see supplementary Figs. 21–24. **b** Proposed exchange mechanism for hydroxylated diphenyl ether. **c** Proposed cleavage mechanisms for hydroxylated diphenyl ether.

**The fast outlier: hydroxylated-DPE**. Only cyclohexanol and phenol were observed as the products from ECH of the hydroxylated model, also known as 4-phenoxyphenol (for details, see Supplementary Fig. 9d). Unlike the above three analogues, this substrate not only reacted faster, but also underwent much faster H/D exchange (Fig. 6a) up to the dideuterated material. This rapid exchange likely reflects the activated keto-enol equilibrium involving the phenol's ortho sites. The $D_{1-2}$ sequential growth processes in Fig. 6a appeared similar to classical base-catalyzed phenolic exchange, albeit substantially faster. As reported by Miranda and co-workers, at reflux in $NaOD/D_2O$, it required 24 h to achieve the di-deuteration of the two ortho phenolic protons[52]. Here, at 60 °C without applied current or any inorganic strong base or acid, the catalytic nickel surface achieved 66% ortho D incorporation in 12 h as shown in Fig. 6a, b.

The above rate and labeling site differences strongly suggest that hydroxylated DPE cleaves via a mechanism different from that working in the other three models. As presented in Fig. 6c, we envision a path where the enhanced binding of the keto tautomer of the phenol ring favors the single-ring coordination, allowing rapid elimination of the phenoxide leaving group, which explains the dramatic increase of the cleavage rate. After phenoxide removal, the surface-bound cyclohexadienone would be either rapidly reduced to cyclohexanol or released and rearomatized via tautomerization back to phenol.

Importantly, a comparison of the inhibitory effects of acetone on the cleavage rates of simple DPE vs. hydroxylated-DPE (HO-DPE) provides additional evidence for the qualitatively different, keto-based path for cleavage of the hydroxylated DPE series. As demonstrated, DPE undergoes C-O cleavage via activation of one of its ortho C-H through double-ring adsorption geometry (Fig. 3a, Route I). HO-DPE is proposed to adsorb via the ketonized ring, resulting in the direct breakdown of its C-O ether

bond. Thus, in the acetone inhibition study (acetone as a competing ketone), we expected a stronger inhibitory effect on HO-DPE than on simple DPE. As expected, the inhibitory effect of HO-DPE was significantly stronger than simple DPE (see Supplementary Fig. 58). In the presence of 5% acetone, the cleavage half-life of DPE only changes from ca. 3 h to 4 h. However, for the hydroxylated analogue, the cleavage half-life changes from ca. 10 min to ca. 2 h, more than 10 times slower, consistent with the potent competition of acetone for the ketone-binding sites needed by the phenoxyphenols to cleave.

**Effects of substituent position on cleavage rates**. Not surprisingly, variations in the positions of the above-discussed substituents also exert potent effects on the rates. As shown in Fig. 7a–c, from para to meta and then to ortho, the closer the functional group is to the ether C-O bond, the slower the cleavage rate. Especially for the methoxy and methyl substituents, the ortho-substituted DPEs had very low reactivity. Importantly, this pattern of rate decreases with changing substituent positions was consistent across all three types of substituents.

The above reactivity trends may be attributed to binding and reactivity modulation due to variations in the torsion angles around the respective ether C-O bonds. The closer the R group lies to the ether linkage between the two aromatic rings, the stronger the expected twisting distortion will be. Compared with a mononuclear molecular catalyst[35], binding to the heterogeneous Ni catalyst surface is presumably more sterically demanding, creating a higher barrier to cleavage in those distorted structures. As shown in Fig. 7d, compared with the para-substituted model, the meta methylated/methoxylated diphenyl ether has a slight twist between the rings, which may displace the ortho C-H slightly relative to the catalytic surface. But in the case of ortho

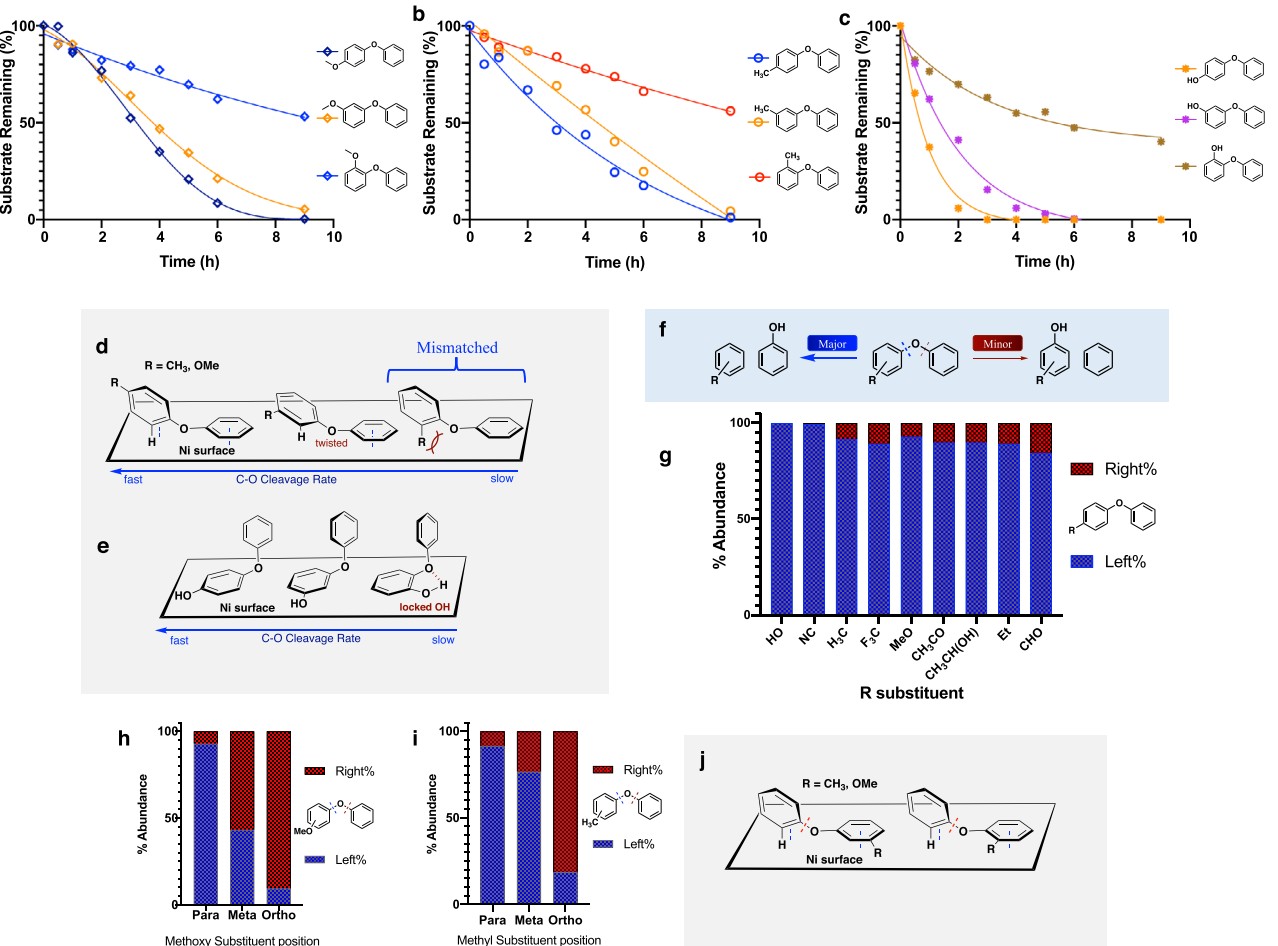

**Fig. 7 Substituent positional modulation of cleavage regioselectivity.** C-O bond cleavage rate comparison of DAEs with different substituents at ortho, meta, and para ring positions under standard ECH condition; 50 mA, 60 °C and pH 8 borate buffer/IPA (2:1): **a** diphenyl ether with a methoxy moiety **b** diphenyl ether with a methyl moiety **c** diphenyl ether with a hydroxy moiety. **d–e** Proposed coordination modes of para, meta and ortho-substituted diphenyl ethers. **f** Regioselectivity of unsymmetrical DAEs. **g** Regioselectivity distribution of different para-substituted diphenyl ethers. Regioselectivity distribution of para, meta and ortho-substituted structures of **h** methoxylated diphenyl ether **i** methylated diphenyl ether. For additional time courses and product analyses see supplementary Figs. 25–27 & 61–67. **j** Switching of binding conformation of meta and ortho-substituted DPE.

substituents, the direct clash between the R group and the catalytic surface generates a mismatch, which significantly inhibits the cleavage. The alternative rotamer would orient the substituent to collide with the π system of the bound ring, also an energetically unfavorable arrangement.

Despite the decreases in cleavage rates with substituent position, for the hydroxylated model, all three structures still react significantly faster than the corresponding methyl and methoxy systems (Fig. 7c). This overall faster conversion supports the idea that the hydroxylated DPE analogues react via mechanisms different from those of the methyl and methoxy cases. Evidently, enhanced substrate binding via enol to keto tautomerization is effective for all three phenoxyphenol isomers, albeit much weaker in the ortho case. We speculate that in the ortho phenoxyphenol (Fig. 7e), intramolecular hydrogen bonding, together with the above torsional effects, may inhibit formation of the keto form that so strongly favors binding and reaction.

**C-O ether bond cleavage regioselectivity in aryl phenyl ethers.**
With one ring substituted, the two aromatic rings are no longer identical, setting up an internal competition between the two possible C-O bond cleavage sites (Fig. 7f). Here we examine the cleavage regioselectivity of the three models (para-substituted

methyl, methoxy, hydroxyl) and several other examples with substituents on the 4 position of one of the benzene rings.

Figure 7g summarizes the regioselectivity distribution of all the tested para-substituted aryl ethers. As shown, all of the ethers preferred cleavage on the left-hand side (blue bars), breaking the C-O bond connected to the substituted aromatic ring. These selectivities are insensitive to variations in current (and thus to electrode potential; for details, see Supplementary Fig. 29). Lowering current only decreases the cleavage rate, presumably due to the decreased rate of active hydrogen production.

Interestingly, whether bearing electron-donating (MeO, OH, CH₃CH(OH)) or electron-withdrawing substituents (CF₃, NC, CHO), this selectivity was observed. However, in these strongly reducing conditions, most of the electron-withdrawing functionalities were reduced prior to aryl C-O cleavage. For example, the acetyl group was rapidly reduced to the corresponding alcohol, which then underwent C-O ether bond cleavage (Supplementary Fig. 61). This result accords with the earlier noted strong preference of the Ni to bind and reduce carbonyl compounds. But even strong bonds like the sp³ C-F in trifluoromethyl were completely reduced to C-H before the C-O ether bond cleavage. No (trifluoromethyl)benzene (the direct cleavage product) was observed, but both toluene and phenoxytoluene, the defluorinated substrate, were easily detected (Supplementary Fig. 66). Thus, all

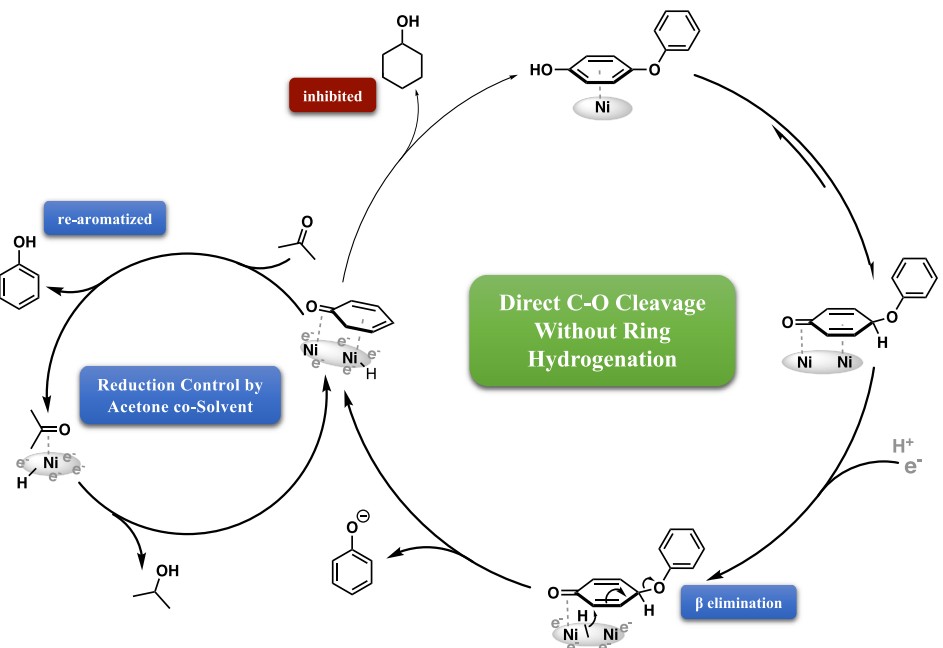

**Fig. 8 Selective inhibition of aromatic ring reduction by acetone co-solvent.** Schematic illustration of product selectivity control by the inclusion of acetone as a co-solvent during ECH of 4-phenoxyphenol.

the fluorines were replaced prior to C-O cleavage. This result contrasts with the oxidative insertion cleavage behavior shown by Hartwig's molecular catalyst system, which selectively broke the C-O bond without removing any benzylic fluorine[37]. Notably, ECH cleavage of the CF$_3$ substituted aryl ether was the slowest among those studied here (for rate comparisons, see Supplementary Fig. 70). Presumably, this slowness reflected the initial diversion of reducing equivalents required to cleave the three C-F bonds.

In general, neither electronic perturbation via variation of substituents nor modification of the applied current (and thus potential) significantly affected cleavage regioselectivity; they did, however, impact cleavage rates, presumably due to additional specific interactions between the substituents and the nickel catalyst.

**Substituent position is the main factor controlling cleavage regioselectivity.** Variation of para substituents from electron-withdrawing CF$_3$ and CN groups to the -OMe donor moiety, led to only minor changes in cleavage regioselectivity. Substituent position was much more important; for both the methoxy and the methyl-substituted systems, the closer the R group was to the C-O bond, the more the cleavage preferred to release benzene and the substituted phenol (Fig. 7h, i). The strong selectivity effect in the ortho-substituted cases was expected; as illustrated in Fig. 7d, an R group attached to the ortho position would directly clash with either the unsubstituted ring, or the catalyst surface if double-ring coordination occurred with the unsubstituted phenyl lying on the surface. Furthermore, with the substituent occupying the ortho site, the nickel-activated benzyne-forming phenol elimination is blocked. However, binding by the substituted ring, a much less favorable interaction, does bypass this steric clash (Fig. 7j), resulting in release of substituted phenol and benzene, but much slower reaction.

As with the para-substituted substrates, isotope exchange experiments were conducted with the meta and ortho analogues. For the phenoxyanisoles, the fastest-reacting para congener showed only slow isotope exchange (Fig. 5a). In the more twisted

meta and ortho cases, the deuterium incorporation was barely detectable (Supplementary Fig. 41). For the phenoxytoluene series, benzylic H/D exchanges in the meta and ortho methyl sites were easily detected (Supplementary Fig. 32), with the less sterically hindered meta isomer showing faster exchange, comparable to that seen in the para case. However, in the ortho phenoxytoluene, benzylic exchange was drastically slower. Thus, the ortho substitution not only redirected the C-O cleavage, but also dramatically slowed direct Ni activation of the benzylic C-H sites.

The hydroxylated DPE series is a key exception to the above regioselectivity switch as a function of substituent position (Supplementary Fig. 31). Though all series show the slowest reactivity in the ortho-substituted cases, the phenoxyphenols do not show reversed cleavage regioselectivity in the ortho-substituted form; like its isomers, the *ortho*-phenoxyphenol gives rise only to phenol and ultimately cyclohexanol products (Supplementary Fig. 27c). This qualitatively different behavior pattern is additional evidence that phenolic substrates react via a mechanism different from the path followed by all the other substituted DAEs studied here.

**Cleavage selectivity is controlled by direct elimination of phenoxide without disruption of aromaticity.** On the basis of the above labeling, rate, and substituent studies, the direct elimination of the leaving group phenoxide (Fig. 3a, Route I) appears most probable as the cleavage pathway for functionalized diphenyl ethers. This scenario controls the C-O bond broken, and therefore product selectivity, in reductive cleavage of diaryl ethers. Given the choice of which ring prefers to adsorb on the nickel surface, it is clear that essentially any substituent, except for -OH, interferes with surface binding.

A particularly emphatic illustration of this idea comes from study of the bis-4-methoxyphenyl ether (Fig. 3d). Although the monosubstituted 4-phenoxyanisole undergoes cleavage at an essentially similar rate as diphenyl ether, this substrate is almost completely unreactive (Fig. 3e). Thus, as depicted, we envision surface binding favoring the unsubstituted phenyl ring adsorbing

**Table 1 co-Solvent Study of Diphenyl Ether: Water Insoluble Substrate.**

| entry | R | co-solv. | conv. | time | yields of products (%) | | |
|---|---|---|---|---|---|---|---|
| | | | | | I | II | III |
| 1 | H | 33% IPA | >99% | 9 h | 13% | 48% | 50% |
| 2 | H | 33% EtOH | >99% | 9 h | 26% | 47% | 35% |
| 3 | H | 50% IPA | 95% | 9 h | 17% | 64% | 30% |
| 4 | H | 5% acetone 28% IPA | 95% | 12 h | 19% | 95% | 2% |
| 5 | OH | 100% buffer | >99% | 1 h | – | 90% | 82% |
| 6 | OH | 10% IPA | >99% | 2 h | – | 85% | 75% |
| 7 | OH | 10% EtOH | >99% | 2 h | – | 66% | 102% |
| 8 | OH | 33% IPA | >99% | 3 h | – | 117% | 40% |
| 9 | OH | 5% acetone | 97% | 6 h | – | 174% | 4% |

For entries 5–9, yield values over 100% reflect the fact that in the case of the phenoxyphenols, cleavage of one ether forms two copies of the product phenol. Note that compared to its formation rate, the faster evaporation rate of benzene (b.p. 80 °C) made it difficult to retain in the heated (60 °C) polar aqueous system (for evaporation studies, see supplementary Fig. 1c). For additional full-time courses and product rate analyses see supplementary Figs. 56–57.

onto a locally flat Ni surface via its π system, placing the ring plane parallel to the surface. Such flat π-bound adsorption geometry has been observed via HREELS in single-crystal adsorption studies, and supported by the computational results of Della Site, Lercher, and co-workers[53,54]. Analogous, albeit highly constrained, structures have been observed in molecular complexes as well[55,56]. This positioning leaves the substituted aryl ring roughly perpendicular, but with one of its ortho C-H sites oriented toward the surface. Binding in this manner suggests that C-O cleavage via oxidative insertion by nickel is unlikely, as the C-O bond cleaved is the one not lying on the surface, and therefore poorly positioned for direct insertion.

Further support for the proposed Route I pathway is found in the isotopic labeling in the products of the ether cleavage (for details and discussion, see Supplementary Figs. 3, 10 and 15). Specifically, the benzene, anisole, and toluene formed show essentially fixed ratios over time of di-, mono-, and undeuterated products (D incorporated on the benzene ring after elimination of the phenoxide). Importantly, the benzyne moiety is a potent ligand for Ni and other metals. Benzynes have been observed for over 30 years in structurally characterized molecular complexes with single Ni centers[57,58] and small Ni clusters[59–62]. Benzyne intermediates have also been identified in LEED and HREELS experiments on metal surfaces, including Ni, that have been dosed with benzene and warmed, with evidence pointing to the benzyne ring plane being tilted from the surface[63]. However, though benzyne-Ni complexes have received some attention as participants in homogeneous processes, there appears little discussion of their possible roles in heterogeneous catalytic reactions[64].

**Selective inhibition with acetone allows selectivity for cyclohexanol or phenol products.** Since the cleavage takes place without immediate reduction of the aromatic products, it is possible to select for phenolic or cyclohexanol products via the use of acetone to prevent phenol reduction, as exploited in the opening discussion that ruled out the formation of cyclohexyl phenyl ether. To illustrate this idea, the para-hydroxylated DPE (4-phenoxyphenol) was chosen, as it is uniformly and rapidly cleaved to two phenol fragments (Fig. 8), albeit by a mechanism clearly different from the above vicinal elimination path.

As phenol is produced by phenoxyphenol cleavage, its reabsorption is outcompeted by acetone adsorption, blocking the sites that otherwise convert phenol to cyclohexanol. Present throughout the reaction as a co-solvent, the acetone slows but does not shut down the (also keto-mediated) cleavage of the ether C-O bond as completely as it does the phenol hydrogenation. A major difference between the two processes could be the energy barrier for reaction; cleavage of the phenoxide only requires one $H_2$ equivalent and is favored by an increase in entropy. However, reduction of bound phenol requires transfer of three $H_2$ equivalents, and entails breaking the aromaticity and strong surface binding of the phenol ring. The practical observation is that acetone selectively inhibits phenol monomer reduction, allowing intentional control over the choice of saturated or aromatic monomer.

**Catalytic method optimization studies.** Any practical process to convert substrates as diverse as biomass and persistent organic pollutants (POPs) must meet four criteria: it must tolerate water, run at modest temperatures and pressures, and be composed of relatively low-cost materials. Considering the basic challenge of cleaving the strong diaryl ether bond, a process offering control of product selectivity would be especially valuable, not only to upgrade or mitigate these feedstocks, but also to convert them to simple mixtures of small-molecule products. Regardless of high laboratory selectivities, a water-sensitive molecular catalyst that requires inert atmosphere conditions and dry organic solvents is of limited applicability. Use of water as the "greenest" solvent would actually be ideal, but most diaryl ethers are poorly soluble in water. Therefore, with the lessons from the above mechanistic analyses and from our previous work cleaving aliphatic-aromatic ethers, we sought to optimize and generalize our ECH method for mild and selective cleavage of water-soluble and -insoluble diaryl ethers.

**Selectivity control via organic co-solvent.** To achieve the cleavage of insoluble diphenyl ethers, a water-miscible organic co-solvent was required. Methanol, ethanol, 2-propanol and acetone were tested as co-solvents to aid in solubilizing and cleaving the diphenyl ether. Due to this substrate's high hydrophobicity, even at high co-solvent/buffer ratios, neither methanol nor ethanol could achieve a homogeneous liquid phase at 60 °C. The more

**Table 2 Substrate Scope of Diphenyl Ether ECH Cleavage.**

| entry | R | R* | time (h) | conv. (%) | $t_{1/2}$ (min) | yields of products (%) | | | | |
|---|---|---|---|---|---|---|---|---|---|---|
| | | | | | | I | II | III | IV | V |
| 1 | Me | Me | 9 | 99 | 219 | 3% | 34% | 45% | 35% | 8% |
| 2 | OMe | OMe | 9 | >99 | 200 | 5% | 48% | 42% | 67% | 2% |
| 3 | CHO | $CH_2OH$ | 9 | >99 | 257 | 9% | 50% | 18% | – | – |
| 4 | $COCH_3$ | $CH(OH)CH_3$ | 9 | 98 | 295 | 8% | 51% | 15% | 76% | 5% |
| 5 | CN | $CH_2NH_2$ | 3 | >99 | 70.0 | 1% | 87% | 3% | – | – |
| 6 | $CF_3$ | $CH_3$ | 9 | 90 | 774 | 5% | 20% | 16% | 20% | 5% |
| 7 | F | H | 6 | >99 | 189 | 13% | 55% | 48% | – | – |
| 8 | Et | Et | 9 | 96 | 234 | 1% | 26% | 49% | 14% | 9% |
| 9 | OH | OH | 3 | >99 | 76.0 | – | 117%* | 40% | – | – |
| 10 | $CH(OH)CH_3$ | $CH(OH)CH_3$ | 9 | 98 | 419 | 9% | 34% | 18% | 50% | 1% |

The electron-withdrawing substituents R undergo fast reduction to R* prior to ether C–O cleavage; for additional quantitative details of intermediates see Supplementary Figs. 61–67. The estimated average current efficiency for the listed diaryl ethers under standard conditions is 22% (± 2%). *See Table 1 for explanation of yields >100%.

amphiphilic 2-propanol, however, at a 1:2 alcohol:buffer ratio (i.e. 33% IPA by volume), was found to be optimal, and was used as the standard reaction co-solvent (2-propanol:pH 8 borate buffer with a 1:2 ratio) for both substrate mapping and H/D exchange experiments. Solutions with less than 33% 2-propanol remained inhomogeneous, and at 50% (Table 1, Entry 3), the higher electrical resistance slowed the cleavage rate (for rate comparison plots, see Supplementary Fig. 56d). A further advantage of using 2-propanol is that it helps maintain a low surface oxide level on the nickel electrode, extending the life and reusability of the catalytic electrode[65]. In higher temperature classical nickel-catalyzed reductions, 2-propanol has been used as a liquid source of hydrogen[66,67].

Though 2-propanol was identified as the generally optimal co-solvent for mild electrocatalytic cleavage of water-soluble DAEs, it was not always the fastest. For water-soluble hydroxylated substrates, introducing the organic co-solvent modestly slowed the rates of both the cleavage (Table 1, Entries 6–8) and phenol hydrogenation, compared with 100% buffer (Entries 5, 8).

The above solvent mixture enables smooth catalytic breakdown of both polar and non-polar diphenyl ethers. But product selectivity can also be tuned (Entries 4, 9) as noted above by adding only 5% v/v of acetone to inhibit the hydrogenation of phenols to cyclohexanols. At higher acetone concentration (10% v/v), ether bond cleavage continued without phenol reduction to cyclohexanol (Supplementary Fig. 57). Typically, even at low potential and current values, phenol reduction is fast, leading always to substantial amounts of phenol ring saturation. With the acetone additive, however, both water-soluble and insoluble diaryl ethers can be selectively cleaved into aromatic monomers with minimal ring reduction. As illustrated in Fig. 8, we envision acetone as a competitive inhibitor, preventing phenol binding in its active keto form but still allowing ether C–O cleavage.

Reticulated vitreous carbon (RVC) was also tested as an electrode for the cathodic electrolysis of diphenyl ether under the same standard ECH condition (50 mA, 60 °C and pH 8 borate buffer/IPA (2:1)). Clearly observed bubbling on the carbon electrode surface suggested successful hydrogen evolution, but essentially no aryl ether cleavage product was detected (Supplementary Fig. 60). This indicated that simple cathodic electron transfer is not capable of cleaving these strong aryl ether bonds; the Ni catalyst is required to activate the arene rings for C–O

cleavage. The critical role of the Ni catalyst was also confirmed in the above H/D exchange studies, which used the same skeletal Ni catalysts.

**Broadening the scope and applications of skeletal Ni ECH: Hydrodefluorination.** As summarized in Table 2, a range of different substituted DAEs was examined via skeletal Ni electrode catalyzed ECH under standard conditions (ambient pressure, 60 °C, 50 mA). The estimated average faradaic efficiency of the listed diaryl ethers is 22% (± 2%). As reported before[49], however, both the catalyst pre-equilibration by applying current prior to substrate injection, and the etching of the Al during the catalyst preparation introduce some reducing power in the cathode, which complicates the calculation of pure faradaic yields. Most significantly, the skeletal Ni electrode was also able to cleave not only DAE's with classic donor substituents, but also those with electron-withdrawing functionalities (Table 2, Entries 3–7). Interestingly, as mentioned above, hydrodefluorination of 4-fluoro- and 4-trifluoromethyl diphenyl ethers was observed, highlighting the ability of the skeletal Ni electrode to activate carbon-fluorine bonds (both $sp^2$ C-F and $sp^3$ C-F), some of the strongest single bonds in organic chemistry. Nickel-catalyzed hydrodefluorination of a fluorophenol at high $H_2$ pressure had been previously reported[68], but to our knowledge, this is the first report of electrocatalytic hydrodefluorination of a perfluoroalkyl substituted diphenyl ether. This finding opens the door to the use of electrodes made from earth-abundant nickel to catalyze aqueous-phase defluorination of aliphatic C-F sites, suggesting a green strategy for PFAS mitigation, as well as potentially useful organic transformations.

**Discussion**

A mild aqueous electrocatalytic hydrogenation/hydrogenolysis (ECH) method has been described for selective cleavage of the diaryl ether moiety, catalyzed by skeletal nickel electrodes, to form arene and phenol products. Patterns of reaction rates and cleavage selectivity between the two ether C–O bonds have been mapped out as a function of substituent structure and position. Together with isotopic labeling studies, two distinct cleavage mechanisms were identified: a direct elimination involving a surface-bound aryne from DPE and its methoxylated and

methylated analogues, and a path involving surface binding of the keto form of the phenoxyphenols, the hydroxylated DPE analogues.

We propose that cleavage of the unsubstituted DPE and its methoxy and methyl congeners occurs via a double-ring coordination mechanism. In this route, the nickel catalyst activates an aromatic C–H site vicinal to the C–O bond, which is then eliminated to release phenoxide and form the surface-bound aryne. Rehydrogenation with surface hydrogen atoms, formed via electrochemical reduction of protons from solution, yields the arene product. Diaryl ethers cleaved via this path were sensitive to changes in substituent position, with cleavage regioselectivity completely switching between para- and ortho-R substitution.

The hydroxylated DPEs are fast outliers, cleaved through a different mechanism which we infer to involve surface binding by the phenol's keto form. This mode significantly enhances the binding of the substituted ring and enables rapid C–O cleavage. Importantly, unlike the dual-ring activation scheme, changes in the hydroxyl position had little impact on the cleavage regioselectivity. Selectivity control via selective inhibition by acetone allows the phenol products to be further reduced to cyclohexanols or retained as valuable aromatics.

This electrocatalytic method is of potential value for reductive fragmentation of both lignin and various persistent organic pollutants (POPs). Importantly, such electrocatalytic strategies are "green" as they enable the coupling of renewable energy (electricity from solar energy) to the processing of sustainable carbon sources (biomass) and the mitigation of pollutants. Having demonstrated effective cleavage of the generally unreactive sp$^2$ ether systems and the activation of C–F bonds, future work will extend these mild, "green" ECH methods to applications in real lignin upgrading and in the remediation of emerging contaminants, while further exploring the scope and functional group tolerance of these processes.

## Data availability

Data (Details of experimental and analytical procedures, isotope labeling procedure and results, full time course measurements, dimer syntheses and characterization via $^1$H NMR spectra) that support the findings of this study are available within the article (and its Supplementary Information files) and from the corresponding authors on request.

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

## Acknowledgements

This material is partly based upon work supported by the U.S. Department of Energy, Office of Science, Office of Biological and Environmental Research under Award Number DESC0018409 (E.L.H. and J.E.J.), work funded by the DOE Great Lakes Bioenergy Research Center (DOE BER Office of Science DE-FC02-07ER64494), and work supported by the National Science Foundation under Grant No. 1603347 (C.M.S. and J.E.J.). C.M.S. acknowledges support from the USDA National Institute of Food and Agriculture, Hatch project 1018335, and Michigan State University AgBioResearch.

## Author contributions

Y.Z. and J.E.J. conceived the study and wrote the paper. Y.Z. performed the experiments and analysed the data. G.E.K. provided samples and analytical assistance, E.L.H. and C.M.S. read and provided useful comments on the manuscript.

## Competing interests

The authors declare no competing interest.
