## [Peer review file · Nature Communications]

REVIEWER COMMENTS

Reviewer #1 (Remarks to the Author):

This manuscript is a beautifully written and clear study of electrocatalytic hydrogenation of diaryl ethers on skeletal Ni. The area of study, catalytic C-O cleavage for the processing of green precursors like biomass is of intense interest. In this work, the authors present a very mild and practical approach to selective C-O cleavage without arene hydrogenation. The authors incorporate well-designed mechanistic studies in the opening section, showing with maximal detail what is occurring by careful studies and logical choice of substrate. The conclusions are convincing and unexpected, but even should future studies prove certain aspects untrue, the data provided here will prove invaluable in refining the mechanism. The paper doesn't stop at mechanistic insight, but also attacks the practical application and scope.

I believe this work aligns itself as a seminal paper in an area of intense current interest and is worthy of publication with very minimal changes, if any.

Minor comments:

Sometimes the language of the paper is different from what I am used to as an organometallic chemist. In the paper, sometimes oxidative addition of a C-H bond is referred to as insertion (which it is from the substrates point of view, but is not the language I am used to seeing for such a reaction). Potential changes to this are at the author's discretion.

However, on at least one occasion the language is potentially confusing:

"This suggests that upon the activation of the benzene ring, forward reaction to cleave the C-O ether bond dominates over reversal that would scramble the ortho hydrogens with surface deuterium atoms.

By using the word activation, do the authors mean coordination to the Ni surface? The word activation is ambiguous here, and could mean cleavage of the C-H or C-O bond, though neither seems to make sense.

Reviewer #2 (Remarks to the Author):

The manuscript "Electrocatalytic Hydrogenolysis (ECH) of Diaryl Ethers Over Skeletal Ni: C-O Cleavage Occurs by Direct Elimination via Benzyne Intermediates" provides an analysis of the mechanism of electrocatalytic C-O cleavage over Ni electrode as model compounds for conversion of lignin. This work is a continuation of the recent publication of authors in J. Am. Chem. Soc. 2020, 142, 8, 4037–4050 focusing on Aryl Alkyl Ether cleavage. The authors performed a deep analysis of the mechanism of the reaction using labeled technique and differently substituted diaryl ethers with identification of the mechanism of C-O cleavage. The work is well done and the conclusions seem to be reasonable. However, some points have to be clarified before acceptance of the manuscript:

1. The manuscript starts directly with an analysis of the products but I would strongly recommend to provide a general scheme of electrical cell and processes over electrodes for non-specialists in electrochemistry.
2. The authors use the same conditions for electrocatalysis (50 mA, 60 °C and pH 8 borate buffer/IPA (2:1)). It would be important to clarify the role of these factors in C-O cleavage reaction. So, what would happen at higher current and temperature...
3. It is hard to understand Figures 1,3 and Tables 1,2 of the manuscript because the total yield of the products is significantly higher than 100 % and the yields of the products do not fit the reaction. For example, in Figure 1b the yield of phenol is close to 100 % and benzene about 20 %. Of course, the high evaporation rates of the chemicals complicate the analysis of the products. However, it would be important to perform analysis correctly with the total yield of the products close to 100 %. The authors could use more efficient condensation of the products or perform continuous analysis of the gases by GC.
4. The generation of significantly higher excess of phenol derivatives in comparison with benzene derivatives products (Table 1,2) besides evaporation could be assigned to the partial C-O cleavage by water with the generation of two phenol species from ether. The labeled water H₂O¹⁸ could help in clarification of this point.
5. The authors explain the high activity of C-O cleavage in aryl ether with OH group on the phenolic ring by equilibrating of the hydroxyl with the keto form. I would say this statement requires additional evidence. These species should be visible by analysis of the electrode surface using FTIR spectroscopy or analysis of the transformation of model ether compound with carbonyl group.
6. The role of acetone in the suppression of phenol hydrogenation implies the hydrogenation of acetone to isopropanol. However, it's not clear what is the rate of acetone hydrogenation in comparison with C-O cleavage. It would be important to show it.

7. The side process of the C-O cleavage reaction is water splitting to H₂. It is not clear what is electrocatalytic efficiency (faradaic yields) of C-O cleavage in comparison with pure water splitting. It is necessary to provide these numbers.

REVIEWER COMMENTS

Reviewer #1 (Remarks to the Author):

This manuscript is a beautifully written and clear study of electrocatalytic hydrogenation of diaryl ethers on skeletal Ni. The area of study, catalytic C-O cleavage for the processing of green precursors like biomass is of intense interest. In this work, the authors present a very mild and practical approach to selective C-O cleavage without arene hydrogenation. The authors incorporate well-designed mechanistic studies in the opening section, showing with maximal detail what is occurring by careful studies and logical choice of substrate. The conclusions are convincing and unexpected, but even should future studies prove certain aspects untrue, the data provided here will prove invaluable in refining the mechanism. The paper doesn't stop at mechanistic insight, but also attacks the practical application and scope.

I believe this work aligns itself as a seminal paper in an area of intense current interest and is worthy of publication with very minimal changes, if any.

Minor comments:

Sometimes the language of the paper is different from what I am used to as an organometallic chemist. In the paper, sometimes oxidative addition of a C-H bond is referred to as insertion (which it is from the substrates point of view, but is not the language I am used to seeing for such a reaction). Potential changes to this are at the author's discretion.

However, on at least one occasion the language is potentially confusing:

"This suggests that upon the activation of the benzene ring, forward reaction to cleave the C-O ether bond dominates over reversal that would scramble the ortho hydrogens with surface deuterium atoms.

By using the word activation, do the authors mean coordination to the Ni surface? The word activation is ambiguous here, and could mean cleavage of the C-H or C-O bond, though neither seems to make sense.

Response: Fixed, with clearer terminology now used: "This suggests that following Ni insertion into the ortho C-H bond of the activated benzene ring, forward reaction..."

Reviewer #2 (Remarks to the Author):

The manuscript "Electrocatalytic Hydrogenolysis (ECH) of Diaryl Ethers Over Skeletal Ni: C-O Cleavage Occurs by Direct Elimination via Benzyne Intermediates" provides an analysis of the mechanism of electrocatalytic C-O cleavage over Ni electrode as model compounds for conversion of lignin. This work is a continuation of the recent publication of authors in J. Am. Chem. Soc. 2020, 142, 8, 4037–4050 focusing on Aryl Alkyl Ether cleavage. The authors performed a deep analysis of the mechanism of the reaction using labeled technique and

differently substituted diaryl ethers with identification of the mechanism of C-O cleavage. The work is well done and the conclusions seem to be reasonable.

However, some points have to be clarified before acceptance of the manuscript:

1. The manuscript starts directly with an analysis of the products but I would strongly recommend to provide a general scheme of electrical cell and processes over electrodes for non-specialists in electrochemistry.

Response: An electrocatalytic hydrogenolysis/hydrogenation scheme has now been added in the introduction (Scheme 3).

2. The authors use the same conditions for electrocatalysis (50 mA, 60 °C and pH 8 borate buffer/IPA (2:1)). It would be important to clarify the role of these factors in C-O cleavage reaction. So, what would happen at higher current and temperature...

Response: Standard reaction conditions (50 mA, 60 °C and 2:1 buffer/IPA) were chosen based on preliminary exploration of these variables. Higher temperatures led to faster evaporation of volatile non-polar products (such as benzene, toluene). As shown in the SI (page S27, Figure S30 d), lower temperatures led to inhomogeneity of the solution, which then lowered the overall conversion and complicated analyses.

The 2:1 borate buffer to IPA ratio was the result of optimizations discussed in the methodology section; we found that lower IPA concentration led to liquid phase inhomogeneity, while higher IPA content slowed the reactions. Other small alcohols were not as effective at promoting solution homogeneity, while maintaining useful reaction rates.

Current variations were also studied (SI page S27, Figure S30 a-c), demonstrating that reaction selectivities do not materially change, though lowering the current slows the overall reaction rate, presumably due to slower proton reduction to the active hydrogen available on the catalyst.

3. It is hard to understand Figures 1,3 and Tables 1,2 of the manuscript because the total yield of the products is significantly higher than 100 % and the yields of the products do not fit the reaction. For example, in Figure 1b the yield of phenol is close to 100 % and benzene about 20 %. Of course, the high evaporation rates of the chemicals complicate the analysis of the products. However, it would be important to perform analysis correctly with the total yield of the products close to 100 %. The authors could use more efficient condensation of the products or perform continuous analysis of the gases by GC.

Response: Figures 1 and 3 and Tables 1 and 2 show the yields of the products. The values summing to over 100% reflect the fact that in the case of each diaryl ether, cleavage of one ether forms two products. For example: cleavage of diphenyl ether forms benzene and phenol (Figure 1) & cleavage of 4-methoxy diphenyl ether forms anisole and phenol (Figure 3). Each of these products is formed in 1:1 proportion from the correspondingly consumed starting material, so their yields should not be summed. Unfortunately, due to its fast evaporation, we were not able to achieve full capture of the

benzene product, despite substantial efforts at improving condensation efficiency. Instead, we performed analysis of the volatile product evaporation rates as given in the SI (page S5, Figure S1). Under the reaction conditions, the evaporation rate of benzene is faster than its formation rate, making it hard to capture in the aqueous buffer solution. In Table 1, entries 5-9, the explicit yield values >100% reflect the fact that phenoxyphenols yield two phenol molecules upon cleavage so that ideally, the phenol and their downstream cyclohexane products would sum to 200%. This is now explained in the caption of Table 1.

4. The generation of significantly higher excess of phenol derivatives in comparison with benzene derivatives products (Table 1,2) besides evaporation could be assigned to the partial C-O cleavage by water with the generation of two phenol species from ether. The labeled water H₂O¹⁸ could help in clarification of this point.

Response: Small amounts of direct hydrolysis of the aryl ether C-O bond cannot be absolutely eliminated, though diaryl ethers are famously recalcitrant to breakdown, as noted in the introduction, and our reaction conditions are exceptionally mild (60 °C). Thermochemically, that cleavage is almost exactly thermoneutral, so there is little driving force for such a reaction. Most straightforwardly, (a) with substituted diaryl ethers such as 4-anisyl phenyl ether (substrate in Figure 3), if the C-O cleavage proceeded via hydrolysis, we would expect formation of 4-methoxyphenol and phenol, but it was anisole and phenol that were the observed cleavage products (with the minimal (~2%) formation of 4-methoxyphenol matched with benzene). Analogous observations apply to the phenoxyphenols and the bis(4-anisyl) ether. This point is now further explored in the manuscript (Page 7, Line 8-10). Note, anisole losses due to evaporation are analyzed in the evaporation study in the SI (page S5). Also (b) catalyzed or uncatalyzed hydrolysis would have been obvious in our catalyst- and current-free controls where two equivalents of phenol would have been formed from each one equivalent of starting diphenyl ether. Catalyzed hydrolysis would presumably be independent of current as it is not a redox process, but we only saw significant cleavage with current flowing.

We appreciate the reviewer's labeling suggestion, but note that one danger in using ¹⁸O water is that ketones readily exchange oxygen with water, and we believe that the phenol reductions proceed through surface-bound ketone forms, as evidenced by ready catalyzed H/D exchange in the ortho and para positions of phenols.

5. The authors explain the high activity of C-O cleavage in aryl ether with OH group on the phenolic ring by equilibrating of the hydroxyl with the keto form. I would say this statement requires additional evidence. These species should be visible by analysis of the electrode surface using FTIR spectroscopy or analysis of the transformation of model ether compound with carbonyl group.

Response: True; ideally, spectroscopy could probe for the surface-bound species. That would be an excellent study complementing this work. However, we do not have the specialized capabilities required for infrared studies on an opaque, extremely rough surface under liquid aqueous medium. Such a study would be a major undertaking as

confirmed by discussions with an electrochemist colleague who commonly uses SERS to probe smooth surfaces (dry or under organic solvents).

The main evidence we invoke for the phenol bonding being ketone-like rests on the large difference in inhibition rates between reactions of phenolic diaryl ethers (e.g. 4-phenoxyphenol) vs. simple diphenyl ether or 4-phenoxyanisole. The discussion of this disparity has now been moved up to page 10 (line 6) to highlight its mechanistic significance. The H/D exchange studies also note much more rapid deuterium incorporation into phenols under our catalytic conditions than aromatics with other substituents.

Regarding diaryl ethers bearing other carbonyl functional groups (-CHO and -COCH₃), the aldehyde or ketone carbonyl groups are reduced in <1 h; see details in SI Figures S61 and Figure S64. This is consistent with the expected high binding affinity and reactivity of carbonyl moieties.

6. The role of acetone in the suppression of phenol hydrogenation implies the hydrogenation of acetone to isopropanol. However, it's not clear what is the rate of acetone hydrogenation in comparison with C-O cleavage. It would be important to show it.

Response: True, acetone can be reduced by ECH to IPA in water [see review in Li et al., Energies 2018, 11, 2691] and specifically at Raney Ni [see X. DeHemptinne, Acad. R. Belg. Cl. Sci. Mem. 1961, 32, 5.]. However, none of this paper's conclusions rest on the extent of this reaction. As discussed above, acetone's role in this work was to control the product selectivity by suppressing phenol reduction, presumably via competition for surface binding. This effect enables the choice to stop at the more valuable phenol or further reduce it to cyclohexanol. In terms of analyses, because the buffer mixture already includes isopropyl alcohol (IPA), analysis for the small amounts of additional IPA formed via acetone reduction is challenging. In the aqueous, salt-containing buffer mixture, quantitating IPA leads to severe peak tailing in the GC-MS chromatograms.

7. The side process of the C-O cleavage reaction is water splitting to H₂. It is not clear what is electrocatalytic efficiency (faradaic yields) of C-O cleavage in comparison with pure water splitting. It is necessary to provide these numbers.

Response: An estimated value for the average faradaic efficiency (~22%) is discussed in the main text of methodology section now (page 16 and the caption of table 2), along with caveats due to catalyst pre-equilibration and the original Raney Ni etching processes. As with acetone reduction discussed above, none of the paper's mechanistic conclusions depend on the quantitative faradaic efficiencies of the ECH.

REVIEWERS' COMMENTS

Reviewer #2 (Remarks to the Author):

The authors provide minor revisions of the manuscript without additional experiments to confirm the mechanism. Generally, I am satisfied with the quality of the manuscript and it could be published in the present form.

REVIEWER COMMENTS

Reviewer #2 (Remarks to the Author):

The authors provide minor revisions of the manuscript without additional experiments to confirm the mechanism. Generally, I am satisfied with the quality of the manuscript and it could be published in the present form.

Response: We thank the referee for their thoughtful and positive review.